# Mechanisms for destabilisation of RNA viruses at air-water and liquid-liquid interfaces

C. A. Brackley [1], A. Lips[1], A. Morozov[1], W. C. K. Poon[1] & D. Marenduzzo [1✉]

Understanding the interactions between viruses and surfaces or interfaces is important, as they provide the principles underpinning the cleaning and disinfection of contaminated surfaces. Yet, the physics of such interactions is currently poorly understood. For instance, there are longstanding experimental observations suggesting that the presence of air-water interfaces can generically inactivate and kill viruses, yet the mechanism underlying this phenomenon remains unknown. Here we use theory and simulations to show that electrostatics may provide one such mechanism, and that this is very general. Thus, we predict that the electrostatic free energy of an RNA virus should increase by several thousands of $k_B T$ as the virion breaches an air-water interface. We also show that the fate of a virus approaching a generic liquid-liquid interface depends strongly on the detailed balance between interfacial and electrostatic forces, which can be tuned, for instance, by choosing different media to contact a virus-laden respiratory droplet. Tunability arises because both the electrostatic and interfacial forces scale similarly with viral size. We propose that these results can be used to design effective strategies for surface disinfection.

---

[1] SUPA, School of Physics and Astronomy, The University of Edinburgh, Peter Guthrie Tait Road, Edinburgh EH9 3FD Scotland, UK. ✉email: dmarendu@ph.ed.ac.uk

The physics of virus-surface and virus-interface interactions is ripe with interesting experimental observations[1,2], but several of these lack a satisfactory theoretical understanding. For instance, the number of viable viruses on a surface typically decays over time as $n(t) = n_0 e^{-t/t_0}$, with the timescale $t_0$ ranging from hours to days[3]. A variety of factors affect $t_0$, such as the nature of the surface and the absolute humidity[4]. However, a mechanistic understanding of this timescale is currently lacking. Equally intriguingly, several reports suggest that viruses are inactivated whenever they are exposed to flow in the presence of air/liquid/solid interfaces, for instance when air is bubbled through a viral solution[5], or when such a solution is tumbled in a test tube[6] or passed through a packed bed of beads[7]. Viral inactivation by exposure to suitable interfaces is clearly potentially relevant to surface disinfection; yet, once again, the physical mechanisms are unknown.

A potentially relevant observation is that viruses are highly charged. This is both because they contain nucleic acids (RNA or DNA) with high negative charge—one electron at each phosphate—and because the proteins constituting their capsid shell have a pH-dependent charge, which can be as high as one electron per $nm^2$ [8–11]. A priori, then, we expect electrostatics to be important to viral energetics. Indeed, the electrostatic energy stored in empty RNA viral capsids is estimated to be $\lesssim 10^4 k_B T$, and electrostatic interactions alone are sufficient to yield spontaneous self-assembly of an RNA virion under physiological conditions[9].

Near an air-water interface, the electrostatic Debye-Hückel repulsion between two point-like charges switches from an exponentially screened interaction to an unscreened and long-range effective dipolar repulsion[12–15], whereas charges wholly in the air phase interact via the Coulomb potential. Thus, electrostatic interactions change fundamentally close to an air-water interface, and the fine balance needed for capsid self-assembly may well be lost under those conditions, potentially leading to viral destabilisation. Indeed, the free energy increase needed to disrupt a virus is likely relatively small, because disassembly is a necessary part of the infection cycle[16]. We, therefore, hypothesize that strong electrostatic interactions at interfaces may lead to viral deactivation.

To test this hypothesis, we solve numerically the non-linear Poisson-Boltzmann (PB) equation for a viral particle approaching an air-water or a liquid–liquid interface. This reveals a significant electrostatic free energy cost opposing adsorption to the interface. We compare this cost with the energy gained when a nanoparticle covers part of an interface, which leads to a saving in interfacial energy and which is responsible for the stabilisation of Pickering emulsions[17,18]. We call this phenomenon the 'Pickering effect' in what follows. Depending on physical parameters such as the dielectric constants and Debye lengths in the two contacting media, we find that the competition between the electrostatic and Pickering effects yields a transition between a regime where the virion breaches the interface spontaneously, and one where it is repelled from it. These findings shed light on previous viral inactivation experiments[5–7], and suggest strategies for effective surface decontamination. Our calculations are distinct from those in previous work, which focussed on solid surfaces rather than fluid interfaces[1].

## Results

### A Poisson-Boltzmann model for an RNA virus close to an interface
In a typical RNA virus, the flexible (negatively charged) RNA is adsorbed to the (positively charged) interior of the protein capsid[9,10]. We model this by two oppositely charged concentric spherical shells of average radius $R$ and with spacing $2\delta$ between them (Fig. 1a). For simplicity, we consider an equal

charge density, $\sigma$, for both shells, so that the viral particle carries a net charge (i.e., it has a non-zero charge monopole). Selected numerical simulations and theoretical arguments (see SI, Supplementary Note 2) suggest that the trends we find are unaffected if the charge densities are tuned to give a neutral virion. We consider a planar interface separating media I and II with inverse Debye lengths $\kappa_1$ and $\kappa_2$ and permittivities $\epsilon_1$ and $\epsilon_2$.

We introduce cylindrical spatial coordinates $z$ (height with respect to the interface plane) and $r$ (perpendicular distance to $z$ axis). The centre of mass of the viral particle lies at $r = 0$ and $z = z_c < 0$. In all our numerical calculations medium I ($z < 0$) is an aqueous physiological buffer, which we model as a 150 mM monovalent salt solution with $\kappa_1^{-1} \sim 1$ nm and $\epsilon_1 \sim 80\epsilon_0$, where $\epsilon_0$ is the dielectric permittivity of vacuum. The capsid interior (medium III in Fig. 1) is assumed here to be in chemical equilibrium with medium I, as the capsid is normally permeable to salt, so we set $\kappa_3 = \kappa_1$ and $\epsilon_3 = \epsilon_1$.

A limitation of our concentric shell model of a virion is that it is highly simplified. The real protein capsid charge is pH-dependent. Moreover, the exterior walls of the capsid tend to be oppositely charged, and enveloped viruses such as influenza are characterised by more complex charge distributions[19]. Including these features is potentially important, and will be of interest to future work. Nevertheless, the system we consider here is a generic RNA virus, and it has been shown that modelling the capsid as a uniformly charged shell gives similar results, as regards PB simulations of the bulk electrostatics, as those obtained by more realistic charge distributions[9]. We also note that we model virions of a fixed shape, which is a good approximation until they are subjected to forces of order of $\sim 1$ nN, at which point capsids may substantially deform, or in some cases may even rupture[20–24].

With monovalent salts in both medium I and II, the non-linear PB equation determining the electrostatic potential of this system, $\phi$, is[25–28]

$$\nabla \cdot \left( \epsilon(r,z) \nabla \tilde{\phi} \right) - \epsilon(r,z) \kappa^2(r,z) \sinh(\tilde{\phi}) = -\frac{e_0}{k_B T} \rho(r,z). \quad (1)$$

Here, $\tilde{\phi} \equiv \frac{e_0 \phi}{k_B T}$ is the dimensionless electrostatic potential, where $e_0$ is the elementary charge and $k_B T$ the thermal energy. We model the charge density of the virion, $\rho(r,z)$, as two oppositely charged shells (see Supplementary Note 1 for the precise functional forms used). For spherically symmetric virions, $\rho(r,z) = \rho(r,z; z_c) = \rho(\sqrt{r^2 + (z - z_c)^2})$ (Fig. 1b; note the parametric dependence on $z_c$). Finally, $\kappa(r,z)$ and $\epsilon(r,z)$ denote the spatially varying inverse Debye length and dielectric permittivity respectively (they also depend parametrically on $z_c$).

The interfacial electrostatics depends on $\kappa_2/\kappa_1$ and the dielectric contrast $\epsilon_2/\epsilon_1$. The importance of non-linear effects is governed by the dimensionless charge density $\sigma^* = \frac{\sigma R e_0}{\epsilon_1 k_B T}$, which compares typical electrostatic and thermal energies[25]. Additional geometrical parameters are $\kappa_1 R$ and $\delta/R$. For an RNA virus with $R \sim 20$–50 nm, and a charge density $\sim 0.1$–0.5 $e_0/nm^2$ in the two shells, $\kappa_1 R \sim 20$–50, and $\sigma^* \sim 10$–100[9,10,29], whereas $\delta/R$ is $\sim 0.1$ is reasonable given molecular sizes of proteins and RNA. We vary $\kappa_2/\kappa_1$ and $\epsilon_2/\epsilon_1$ to model specific interfaces.

The electrostatic self free energy of the system is obtained by adding the self energy of the virion, which is given by the integral of $\frac{1}{2}\rho\phi$ over all space, and the background energy of counterions (see refs. [30,31] and Supplementary Note 1). In our cylindrical

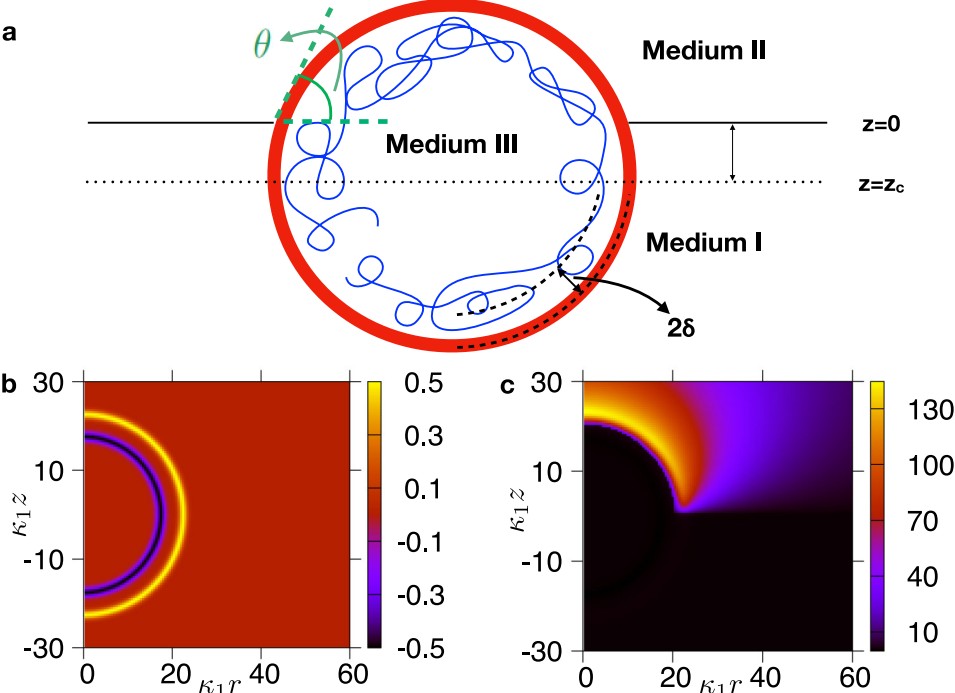

**Fig. 1 System set-up and typical pattern of electrostatic potential. a** Schematics of the system under consideration. The folded RNA and capsid shell of a virion are modelled as two concentric shells with charge density $-\sigma$ and $\sigma$ respectively, a distance $2\delta$ apart. **b** Model charge distribution used in numerical simulations of a virion at an air-water interface ($\kappa_2/\kappa_1 = 0$, $\epsilon_2/\epsilon_1 = 1/80$, $z_c = 0$). The heat map gives the dimensionless local charge density (see SI, Supplementary Note 1). **c** Corresponding potential field found by numerically integrating the non-linear PB equation, Eq. (1), for an air-water interface. The heat map gives the value of the dimensionless electrostatic potential $\tilde{\phi}$.

coordinates, Fig. 1, the system self free energy only depends on $z_c$:

$$\mathcal{F}_{\text{elec}}(z_c) = \pi \int_0^{+\infty} dr \int_{-\infty}^{+\infty} dz \, r\rho\phi$$
$$+ 2\pi \int_0^{+\infty} dr \int_{-\infty}^{+\infty} dz r \epsilon \kappa^2 \left(\frac{k_B T}{e_0}\right)^2 g(\tilde{\phi}) \quad (2)$$
$$g(x) = \frac{x \sinh(x)}{2} - \cosh(x) + 1,$$

as $\rho$, $\kappa$, $\epsilon$ and $\phi$ depend parametrically on $z_c$. We call $\mathcal{F}_{\text{elec}}(z_c)$ the virion-interface 'approach curve'. The electrostatic force exerted by the interface on the virion is given by $f(z_c) = -\frac{\partial \mathcal{F}_{\text{elec}}(z_c)}{\partial z_c}$. We note that the self-energy $\mathcal{F}_{\text{elec}}$ is the energy required to assemble the virion by bringing in its components from infinity. It can not be used to predict whether the self-assembly of a virion is thermodynamically favourable or not: to do so, one needs to subtract the free energy of an empty capsid as in[30]. Nevertheless, this is not an issue here as we are primarily interested in the change in free energy with respect to $z_c$, as the virion approaches the interface.

**Electrostatics provide a general physical mechanism for viral destabilisation at air-water interfaces.** We first compute the self free energy of a viral particle approaching an air-water interface ($\kappa_2 = 0$, $\epsilon_2/\epsilon_1 = 1/80$), by using a relaxation algorithm to solve Eq. (1) (see SI, Supplementary Note 1). Typically we take $\kappa_1 R = 20$, $\delta/R = 0.125$ and $\sigma^* \simeq 17.2$ to be physiologically relevant. Approaching the interface increases the self-free energy due to a generic build-up of electrostatic repulsion arising from the proximity of the air phase where there is no screening, and the permittivity is much smaller (Fig. 2a). This increase is substantial, and is of the order of $10^4 k_B T$ for a viral particle close to the interface.

The electrostatic free energy increase $\Delta\mathcal{F}_{\text{elec}} = \mathcal{F}_{\text{elec}}(0) - \mathcal{F}_{\text{elec}}(z_c \to -\infty)$ opposing adsorption is found numerically to scale as $R^2$ (Fig. 2a, inset), as predicted by an analytic calculation (see below and SI, Supplementary Notes 2, 3 for details). Remarkably, this is the same scaling as van der Waals and hydrophobic interactions, which keep the protein capsid together, and which are also of a similar magnitude as the electrostatic free energy increase. These interactions are also similar in magnitude to the calculated electrostatic free energy increase, and we, therefore, hypothese that the total energy of a virus lodged at an interface may become positive and trigger destabilisation or disassembly. Electrostatics is therefore a generic physical mechanism for viral destabilisation at air-water interfaces.

The energy increases with decreasing $|z_c|$, so that viruses are strongly repelled electrostatically from an air-water interface on approach. However, when $|z_c| < R$, the virus breaches the interface and an additional component of the free energy must be considered. The virion now covers a circular part of the interface of area $\pi(R^2 - z_c^2)$, reducing the total free energy. The corresponding 'Pickering free energy'[17,18]—for a spherical viral particle breaching an interface with surface tension $\gamma$— can be estimated as

$$\mathcal{F}_{\text{Pick}}(z_c) = -\pi\gamma(R^2 - z_c^2), \quad |z_c| \le R$$
$$\mathcal{F}_{\text{Pick}}(z_c) = 0, \quad |z_c| \ge R. \quad (3)$$

For an air-water interface, $\gamma \sim 70$ mN/m, so that the Pickering free energy gain for small RNA viruses with $R \sim 20$ nm is of the same order of magnitude as the electrostatic free energy increase computed in Fig. 2a. Whether the Pickering or the electrostatic contribution wins then depends on parameter details, such as the exact charge density of the virion. For the case considered in Fig. 2a, the minimum of the total free energy $\mathcal{F} = \mathcal{F}_{\text{elec}} + \mathcal{F}_{\text{Pick}}$ occurs at $\kappa_1 z_c \simeq -10$ (Fig. 2b), corresponding to an apparent

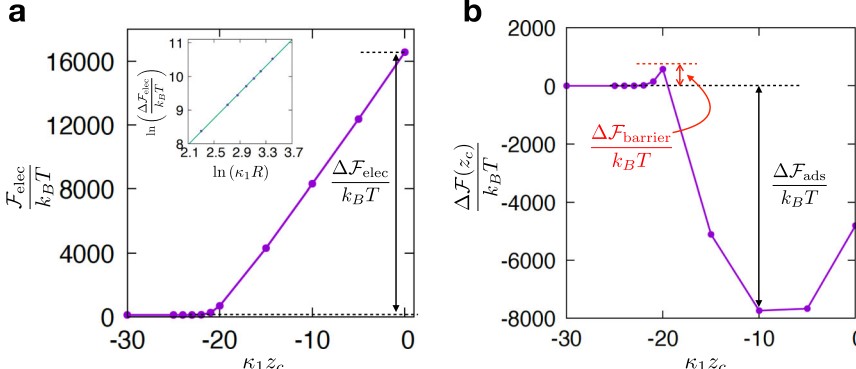

**Fig. 2 Free energy for an RNA virus approaching an air-water interface. a** Approach curve showing the increase in electrostatic free energy $\mathcal{F}_{elec}(z_c)/k_BT$ as a function of distance to the air-water interface. Parameters are as specified in the text. Inset: scaling of the electrostatic free energy increase $\Delta\mathcal{F}_{elec} = \mathcal{F}_{elec}(0) - \mathcal{F}_{elec}(\kappa_1 z_c \rightarrow -\infty)$ in units of $k_BT$ as a function of dimensionless radius of the virion, $\kappa_1 R$. The line is a fit, and has slope ~1.95, which is close to the predicted value of 2, see Supplementary Note 3. **b** Plot of the total (electrostatic plus Pickering) free energy change, $\Delta\mathcal{F}(\kappa_1 z_c) = \mathcal{F}(\kappa_1 z_c) - \mathcal{F}(\kappa_1 z_c \rightarrow -\infty)$, as a virion approaches an air-water interface ($\gamma = 70$ mN/m). The electrostatic free energy barrier $\Delta\mathcal{F}_{barrier}$ and the adsorption free energy gain $\Delta\mathcal{F}_{ads}$ are shown in the plot.

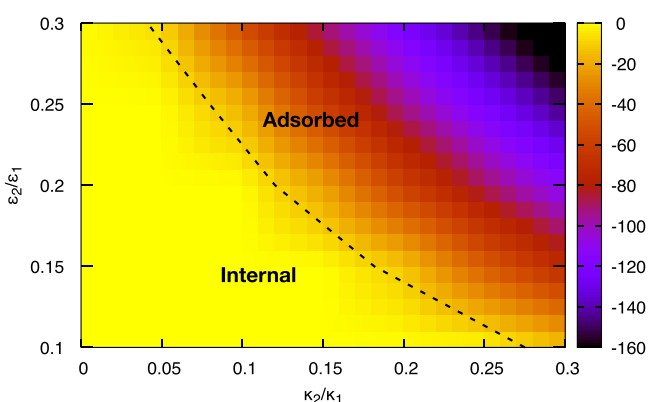

**Fig. 3 Phase diagram for an RNA virus approaching liquid–liquid interfaces.** Phase diagram showing the fate of a viral particle approaching an interface between a physiological aqueous medium and another medium with variable electrostatic parameters. The heat map shows the adsorption free energy, $\Delta\mathcal{F}_{ads}$, in units of $k_BT$, which includes both the electrostatic and Pickering contribution. The dashed line shows the location of the phase boundary between the phase where the system free energy is minimised with the virus in the aqueous phase (internal, $z_c < -R$) or at the interface (adsorbed, $z_c > -R$). Parameters are as in Figs. 1 and 2, except for $\gamma$, which is now 1.5 mN/m to model the lower surface tension of liquid–liquid interfaces.

contact angle (Fig. 1a) of $\theta \sim \pi/3$ (or to $z_c/R \simeq -0.5$). The nontrivial $\theta$ ($\neq 0$ or $\pi/2$) is due to the different scaling of the electrostatic and Pickering free energies with $z_c$: the former is approximately linear (Fig. 2a), the latter quadratic (Eq. (3)).

Even for cases where the Pickering free energy gain is sufficient to favour adsorption (as in Fig. 2), there is an energy barrier opposing this process. This barrier is purely electrostatic because it appears before the virion contacts the interface (when the Pickering contribution is zero), and is given by $\Delta\mathcal{F}_{barrier} = \mathcal{F}_{elec}(-\kappa_1 R) - \mathcal{F}_{elec}(\kappa_1 z_c \rightarrow -\infty)$, which is $\simeq 570 k_BT$ in Fig. 2b. This barrier is too large for Brownian motion to overcome. Interestingly, experiments observe inactivation in viral suspensions following bubbling[5] or tumbling[6], suggesting that the process is not spontaneous, but indeed involves the non-thermal forces overcoming some free energy barrier. Inspection of approach curves such as Fig. 2a reveals that the force resisting adsorption and associated with the electrostatic free energy

barrier is ~0.1–1 nN for typical viral parameters (see, e.g., Supplementary Fig. 2; the repulsion force scales as $R$). To exert a viscous drag of this magnitude in a fluid of viscosity $\eta \sim 1$ cP, a velocity in the range of $v \sim 0.5$–5 ms$^{-1}$ is needed for virions with diameter $R = 20$ nm, which is plausible in vigorous shaking or tumbling.

**Interfacial and electrostatic forces determine the fate of a virion close to a liquid–liquid interface.** Consider now the balance between electrostatic cost and Pickering gain for different liquid–liquid interfaces (for which $\gamma$ is much lower, typically ~1–10 mN/m) as a function of $\kappa_2/\kappa_1$ and $\epsilon_2/\epsilon_1$. This question is of fundamental interest, as the case of $\kappa_2 \neq 0$ was recently shown to be qualitatively different from that of $\kappa_2 = 0$ (relevant for air-water interfaces) as it leads to a distinct interparticle potential at the interface[28,32]. Figure 3 shows the total adsorption free energy $\Delta\mathcal{F}_{ads}$ (defined in Fig. 2b) for $0 \leq \kappa_2/\kappa_1 \leq 0.3$ and $0.1 \leq \epsilon_2/\epsilon_1 \leq 0.3$. The virus switches from being preferentially located in the aqueous phase at low $\kappa_2/\kappa_1$ or $\epsilon_2/\epsilon_1$ due to high electrostatic self energy, to being spontaneously adsorbed at large $\kappa_2/\kappa_1$ or $\epsilon_2/\epsilon_1$ due to the Pickering energy gain.

To qualitatively understand these results, we formulate a Debye-Hückel scaling theory valid for $\kappa_1 R \gg 1$, $\delta/R \ll 1$, and small $\sigma^*$. We estimate $\mathcal{F}_{elec}$ for a virion breaching the interface between fluids I and II as the sum of the self-energies of the two spherical caps in the two media (see SI, Supplementary Note 3), neglecting their interaction, which is numerically smaller because charges on the two halves are typically further apart than charges within each half. In this framework, the electrostatic self-energy of the virion at the interface is

$$\mathcal{F}_{elec}(z_c) \sim \pi\sigma^2 R^2 \left( \frac{1 - e^{-2\kappa_1\delta}}{\epsilon_1\kappa_1} + \frac{1 - e^{-2\kappa_2\delta}}{\epsilon_2\kappa_2} \right)$$
$$+ \pi\sigma^2 R z_c \left( \frac{1 - e^{-2\kappa_2\delta}}{\epsilon_2\kappa_2} - \frac{1 - e^{-2\kappa_1\delta}}{\epsilon_1\kappa_1} \right). \quad (4)$$

Non-linear effects modify both the numerical coefficients and the dependency on $\kappa_1\delta$ or $\kappa_2\delta$ in Eq. (4); our simplified theory also does not capture the presence of a non-zero electrostatic free energy barrier $\Delta\mathcal{F}_{barrier}$ prior to interface contact. Thus, quantitative predictions require full PB numerics (as studied numerically here in Fig. 3). Nevertheless, Eq. (4) can explain qualitatively the results in Figs. 2 and 3.

First, it predicts that $\mathcal{F}_{elec}$ is linear in $z_c$ for all $\kappa_2$ and $\epsilon_2$, matching our numerics in Fig. 2 and Supplementary Fig. 1. Secondly, Eqs. (3) and (4) predict (see Supplementary Note 3) virion adsorption when

$$\sigma^2 \left( \frac{1 - e^{-2\kappa_2\delta}}{\epsilon_2\kappa_2} - \frac{1 - e^{-2\kappa_1\delta}}{\epsilon_1\kappa_1} \right) - 2\gamma \leq 0, \qquad (5)$$

which gives a phase diagram in semi-quantitative agreement with Fig. 3 (see Supplementary Fig. 3). This equation also identifies the electrocapillary numbers $\frac{\sigma^2}{\gamma\epsilon_i\kappa_i}$ (with $i = 1, 2$), which measure the relative importance of electrostatic and interfacial effects in the two contacting media and determine the virion fate at an interface.

To see the relevance of a transition between internal and adsorbed states (Fig. 3) to disinfection, note first that respiratory RNA virions such as severe respiratory syndrome coronavirus 2 (SARS CoV-2) are borne by droplets rich in mucin, a very high molecular weight protein. As such droplets dry on surfaces, the mucin may form a gel state[33] that, as in other hydrogels[34], remains permanently hydrated. When cleaning fluids contact such a composite object, there will be at least a transient interface between liquids of different compositions. As a test case, consider an interface between physiological saline and ethanol, the latter being a common ingredient in hand sanitizers.

In this case, $\kappa_2/\kappa_1 \sim 0.05$, and $\epsilon_2/\epsilon_1 \simeq 0.3$[35], which is close to our predicted transition boundary between 'internal' and 'adsorbed' phases, while staying on the adsorbed side for the parameters used for Fig. 3. Importantly, the electrostatic $\Delta\mathcal{F}_{barrier}$ is over an order of magnitude smaller than that associated with adsorption of the same virion onto an air-water interface (see Supplementary Fig. 1): i.e., the presence of ethanol renders it easier to drive virions close to an interface as the barrier is dramatically reduced. However, there remains enough dielectric contrast for destabilisation through the increase in the electrostatic self-energy term once on the interface. [Indeed, a disassembled virion may be particularly stable at the interface, because the Pickering free energy gain remains, while the large electrostatic free energy cost from charge confinement inside the capsid vanishes.] This may be one reason why alcohol is an efficient disinfectant. A more complete theory would also require modelling the separate effect of ethanol on the genetic material.

A related application is to a virion-laden saline droplet on the skin. The relevant interfaces are now between physiological saline and sebum or sweat. The liquid components of sebum (triglycerides and squalene[36]) have $\epsilon \gtrsim 2\epsilon_0$ and $\kappa \approx 0$[37]. We predict that viruses should stay inside the droplet and not be transferred into sebum. On the other hand, sweat is essentially a salt solution but at somewhat lower concentration than physiological saline[38], so that virions should adsorb at the transient sweat-droplet interface. These predictions have evident implications for viral transmission via touching.

We stress again here that the ability to fine tune interfacial parameters to determine the virus fate depends in large part on the fact that both Pickering and electrostatic contributions scale identically with the particle size, as $R^2$. In addition, it is intriguing that van der Waals interactions, which keep the RNA viral capsid together, are also expected to scale as $R^2$ [9]. It appears therefore that RNA viruses are placed at the edge of thermodynamic stability, which is reasonable for a system that needs to self assemble or disassemble on demand, in response to subtle changes in the surrounding medium. We suggest this very fact, though, makes it possible to design optimal disinfection strategies.

## Discussion

In summary, we have computed the electrostatic free energy of an RNA virus approaching an interface between physiological saline and another medium. Our Poisson-Boltzmann formalism takes into account the spatial charge distribution of the virion and non-linear effects due to the highly charged nature of the virion's constituents. We also provide a simplified Debye-Hückel scaling theory which qualitatively reproduces the trends we observe, whilst being of limited quantitative validity in view of the neglect of non-linear effects. Our main finding is that at an air-water interface the electrostatic energy of a virus increases dramatically, by many thousands of $k_B T$, due to the low permittivity of and the absence of electrostatic screening in air. This fact leads to virion destabilisation, which may be sufficient to trigger viral inactivation or disassembly, thereby providing an appealing physical mechanism to explain longstanding observations of viral inactivation at air-water interfaces[5–7].

Including the Pickering free energy gain, which arises when any nanoparticle covers part of the interface between two media, leads to a transition between a regime where the viral particle is repelled from the interface to one where it adsorbs to it. Focussing on liquid–liquid interfaces, we find that varying the dielectric permittivities and Debye lengths of the two contacting media 'tunes' which regime a given system is in. Potentially, then, disinfectants could, by design, be positioned close to the transition to give a low free energy barrier to adsorption. Virions in droplets in contact with such a cleaning fluid may then adsorb, and therefore deactivate, more easily than when the droplet is in contact with air alone, where external energy is needed to achieve the same end[5–7].

Our predictions are susceptible to experimental testing. Revisiting the classic viral deactivation experiments, e.g., of bubbling air through viral solutions, but now scanning the $(\epsilon_2/\epsilon_1, \kappa_2/\kappa_1)$ parameter space under carefully controlled conditions and detecting viral adsorption at interfaces would be one way forward. It should also be possible to compare the adsorption behaviour of RNA viruses with their empty counterparts that lack the genome[39]. We predict a significant difference in their electrostatic behaviour near interfaces because the charge distribution of the latter is a single charged shell (see Supplementary Note 3).

We have focussed on RNA viruses; but similar considerations should apply to DNA viruses such as bacteriophages, which are also inactivated at interfaces[6]. Yet, there will be quantitative differences, as the DNA of bacteriophages is arranged in a space-filling spool rather than on a thin shell, so that its electrostatic self-energy should differ substantially (see SI, Supplementary Notes 2, 3). As bacteriophages do not self-assemble but use a motor to package their genome, the electrostatic energy increase at the interface may result in DNA ejection rather than capsid disassembly[6,40]. We finally hope to report on generalised theories to consider different charge distributions, charge densities and capsid geometries in the future, as all these parameters are likely important to determine the fate of a virion at an interface. Other interesting avenues are to consider polyvalent counterions in the theory using methods as in ref. [41], or to increase the resolution of the virion model and use a 3D RNA distribution mirroring that of selected viruses of interest, as these are often available experimentally (see e.g.[19]).

**Reporting summary**. Further information on research design is available in the Nature Research Reporting Summary linked to this article.

## Data availability

The datasets generated during and/or analysed during the current study have been deposited in Edinburgh DataShare https://doi.org/10.7488/ds/3152. The datasets generated during and/or analysed during the current study are also available from the corresponding author upon request.

## Code availability

Custom codes written to simulate the Poisson-Boltzmann equation for a model RNA virion close to an interface, and to analyse corresponding data are available from the corresponding author upon request, or they can be downloaded from https://git.ecdf.ed.ac.uk/dmarendu/viral-interface.

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

## Author contributions

C.A.B., A.L., A.M., W.C.K.P. and D.M. designed research; C.A.B., A.M. and D.M. performed research; A.B., A.L., A.M., W.C. K.P. and D.M. analysed the data and wrote the manuscript.

## Competing interests

The authors declare no competing interests.
