## [Peer Review File · Nature Communications]

Mechanisms for destabilisation of RNA viruses at air-water and liquid-liquid interfacesReviewers' Comments:

Reviewer #1:

Remarks to the Author:

In this manuscript Brackley et al. propose an interesting and sound theoretical calculation of the free energy of virus particles at air-water and liquid-liquid interfaces. For this purpose, the authors have numerically solved the Poisson-Boltzmann equation at these interfaces to estimate the free energy of a RNA-virion. I think that the paper is well written and explores an interesting phenomenology never considered before. However, I have a number of questions/suggestions that could improve and clarify some obscure points of the manuscript before considering publication.

In particular, the authors simplify an enveloped virus, such as SARS-CoV-2 as two concentric shells with opposed charge density for RNA (negative) and the virus lipid shell (positive). Actually, in the structure of coronavirus and other RNA viruses, such as influenza A, RNA is assembled in complex ribonucleoproteins structures far away of being a shell (Science 2012, 338 (6114), 1634-1637). In the case of the lipid shell, it is known that the lipid membrane of a cell can be understood as a capacitor whose negative and positive plates are located inside and outside the cell, respectively. However, at the end of page 1, the manuscript reads "... is adsorbed to the (positively charged) interior of the protein capsid". In particular, SARS-CoV-2 is an enveloped virus whose capsid is made of lipids. Enveloped viruses are softer than protein viruses, showing stiffness ten times lower (0.01 N/m) than protein capsids (Biophysical Journal (2011) 100(3) 637-645). In particular, flue is apparently very elastic and does not show fragile rupture under compressing forces. Therefore, it is difficult to compare with the breaking force of protein viruses as proposed by the authors (19). My question here: how can the authors justify that their model is enough to simulate enveloped viruses such as SARS-CoV-2? I think it is a bit of a stretch to pretend that this model can describe SARS-CoV-2. In page two the manuscript reads "...the dimensionless charge density σ^* ..." which depends on the charge density σ . They take σ^* to be about 17.2. How does this value fit with the charge of viruses? Nanoscale, 2015,7, 17289-17298.

In the page three the manuscript reads "... the force resisting adsorption and associated with the electrostatic free energy barrier is $\sim 0.1 - 1$ nN for typical viral parameters." I am not sure that this force can destroy a soft enveloped virus. Even in the case of the fragile protein viruses, 1 nN is in the low limit of their rupture force (PNAS (2004) 101, 7600, PNAS (2009) 106 (24) 9673-9678, J Biol Chem. 2012 Sep 7; 287(37): 31582-31595, eLife 2018;7:e37295. DOI, Nanoscale 2019 11(9):4015-4024))

Technical comments.

In the description of the model (fig. 1a) I miss the separation between the shells 2δ .

It would be helpful to include the force value (N) at the right axis of the charts at figure 2.

Reviewer #2:

Remarks to the Author:

The authors study the interactions between viruses and interfaces using analytical theory and simulations. They find that electrostatics can explain why the presence of air-water interfaces can inactivate and kill viruses. They solve the non-linear Poisson-Boltzmann equation for a viral particle to obtain its behavior at an air-water or a liquid-liquid interface. They find that at an air-water interface the electrostatic energy and in consequence the free energy of RNA viruses increase significantly, due to the low permittivity of and the absence of electrostatic screening in air.

This is an interesting paper because of its relevance to the cleaning and disinfection of contaminated surfaces. The authors prove that electrostatics could be the underlying physical mechanism for viral inactivation at air-water interfaces. The paper is relatively well-written. However, the authors need to clarify several assumptions that they have made throughout the paper. The authors might consider the following suggestions to improve the paper:

1. The authors state that the electrostatic energy stored in viral capsids and genomes is estimated to

be $10^4 k_{BT}$ for typical RNA viruses. It is not clear why this force is positive. Many RNA viruses assemble spontaneously both in vivo and in vitro due to the attractive electrostatics interactions. The electrostatic interaction is indeed the driving force for assembly. I agree that in the case of empty shells, the electrostatics only opposes to the assembly. However, the interaction of RNA and capsid proteins promote the assembly of virus particles. Indeed, in many cases, the viral shells do not assemble in the absence of RNA. The virus mostly assembles due to the interaction of RNA and capsid proteins. The authors need to explain this. Based on the authors discussion, the electrostatic interaction between RNA and proteins should get stronger at the water-air interface due to lack of screening,

2. The authors need to explain what pickering effect is in the introduction where the first time talk about it.

3. The authors assume that the number of charges on the shell is more than the number of charges on RNA. In reality, the number of charges on RNA is way more than those of viral proteins. Does a higher charge density in the inner shell have an impact on the conclusion of the paper? Can the authors solve the problem if the charge densities are different?

4. Most viruses are permeable to salt and water. Is there a reason that the authors consider a different salt concentration inside and outside of a virus at the interface.

5. While calculating the electrostatic interaction, do the authors consider the background energy? This is very important and can change the result of the paper. See for example the work of Podgornik and collaborators: Siber A and Podgornik R 2008 Phys. Rev. E 78 051915 and Erdemci-Tandogan G, Wagner J, van der Schoot P, Podgornik R and Zandi R 2016 Phys. Rev. E 94 022408

Reviewer #3:

Remarks to the Author:

"Electrostatic inactivation of RNA viruses at air-water and liquid-liquid interfaces" is an interesting paper on a timely subject. Overall the ideas promulgated by the authors are new and reasonable but their formal development and implementation is not yet quite convincing. I would be happy to support the publication of this paper once the issues described in detail below are successfully confronted and resolved. They include:

- "we solve numerically the non-linear Poisson-Boltzmann (PB) equation for a viral particle approaching an air-water or a liquid-liquid interface." There is no indication of how they calculated the electrostatic interaction free energy for the full non-linear PB theory. In fact, the electrostatic self-energy, Eq. (2), is actually valid only for the Debye-Huckel approximation and not for the full PB equation, Eq. (1). The correct form of the electrostatic free energy corresponding to the full non-linear PB equation is described in Verwey-Overbeek either with Eq. 23, Eq. 26, in chapter III. If the self-energy expression Eq. 2 was used for all the results described, then they contain crucial errors as one should use the full non-linear PB theory at the specified values of the parameters. This needs to be clearly specified either in the main text of the paper or in the SI.

- "charged concentric spherical shells of average radius R and with spacing 2δ between them (Fig. 1a)." The figure does not indicate what is 2δ . The figure should correspond clearly with the model and should contain explicitly also the 2δ .

- "we consider an equal charge density, σ , for both shells, so that the viral particle carries a net charge". The charge of the virions have been measured in electrophoretic mobility experiments (see several Gelbart et al. papers) and is indeed finite, depending crucially on the pH and ionic strength of the solution. However, contrary to the stated assumption of the paper, there are indications that the inner surface charge makes no contribution to the overall charge of the virus. This is what transpires from, e.g., the AFM studies by Hernandez-Perez, et al. who found that in the case of the adenovirus the presence of its DNA did not affect the overall charge, while experiments by Johnson, et al., for both CCMV and BMV, show that the electrophoretic mobility is insensitive to the packaged RNA. Thus

the assumption of the outer and inner charged shell additively contributing to the total charge of the virus in the paper cannot be clearly substantiated by experiments.

- "The interior of a capsid (medium III in Fig. 1) is likely a different electrostatic environment from the aqueous surrounding. We set $\kappa = 0.1\kappa$ (as the capsid may only be partially permeable to salt) and $\epsilon_3 = \epsilon_1/16$ [9]." The medium III is usually in chemical equilibrium with medium I as the protein-based capsid is permeable to ions while the lipid membrane envelope contains proteins which can act as ion channels. The assumption that the screening in medium I and III differ thus has no ground in actual virus properties. Also, checking the cited Ref [9] confirms this statement, as that paper refers to the dielectric constant of the protein envelope and not the inside medium, which appears to be about 1/16 of the water epsilon. Also the authors do not vary the screening length inside ("The properties of the virion interior, ϵ_3 and κ_3 , also affect the results, but they are not varied below.") so one cannot assess whether this erroneous assumption has any effect on the results or not. This needs to be checked and analyzed, while the assumption of different Debye length inside and outside the virion needs to be dropped.

- the scaling on the size of the capsid, R , is also obtained purely from the linearized DH theory and can be only valid for small values of the surface charge. The scaling needs to be checked also with the full PB theory, unless it can be clearly shown that the regime under consideration fall outside its range of validity.

- the proposed model explicitly ignores pH effects and has thus a very limited validity, as acknowledged in the MS. The conclusions are thus at best suggestive.

- while the authors state that "Electrostatics is therefore a generic physical mechanism for viral inactivation at air-water interfaces" their data (Fig. 2) and the overall description on p.3 seem to indicate rather the reverse, it seems to be the free energy associated with the surface breaching (the Pickering free energy) that favours adsorption. The electrostatics actually opposes it. And quite strongly for that matter so that without the Pickering term the virus is actually strongly repelled from the surface. In this sense the title of the paper is a bit misleading as it is the opposite, i.e., the non-electrostatic forces of the Pickering type, that appear to inactivate the virus. One might suggest the title of the paper should suggest this finding.

- the discussion of the electrostatic forces indicates that they are mostly due to image interactions, either because of the discontinuity of the dielectric properties or the discontinuity in the screening parameter, both lead to a kind of dielectric images, as is known from the literature. Interestingly, the authors never mention any image effects which seem to be crucial for the resulting interactions between a virion and an interface. I think the image mechanism of the strong electrostatic repulsion needs to be clearly analyzed and discussed explicitly.

- the authors state that "To understand these results, we formulate a Debye-Hückel scaling theory valid for $\kappa_1 R \gg 1$ and $\delta/R \ll 1$, which is physically relevant for RNA viruses." There is no mention of the charges in this statement which are crucial for estimating the validity of the DH approximation. The authors actually admit that "quantitative predictions require full PB numerics." and it remains unclear how Eq. 4 could be used even qualitatively. The limits of validity especially those depending on the charge density should be established clearly and explicitly.

- in analyzing the disinfecting properties of ethanol the authors fail to mention the effect of ethanol on the genetical material, which is possibly much larger than the dielectric effects that they are describing. Ethanol (and other alcohols) increase the electrostatic coupling and would promote a condensation of the genome, rendering it ineffective. The authors could comment on this.

- the crucial scaling as R^2 for both the Pickering and the electrostatic parts that is required for the fine tuning the interfacial parameters needs to be viewed with caution as the linearized electrostatic

theory consistent with the R^2 scaling cannot be extended outside its regime of validity. Also, the assumption of a thin RNA shell at the surface is a very idealized depiction of the RNA distribution in the real viruses. If the results described remain valid only for such a thin RNA shell they should be viewed more as an artifact of the idealized model than a real physical property of virus shells or complete virions.

- while the authors explicitly state that "Our Poisson-Boltzmann formalism takes into account the spatial charge distribution of the virion and non-linear effects due to the highly charged nature of the virion's constituents." the bulk of the results depend on the Debye-Huckel calculations and would probably not hold in the full non-linear PB framework. This needs to be addressed and the reader should get a clear idea what are the limitations of approximations and what are the limitations of the model.

Reply to Reviewer #1

COMMENT:

In this manuscript Brackley et al. propose an interesting and sound theoretical calculation of the free energy of virus particles at air-water and liquid-liquid interfaces. For this purpose, the authors have numerically solved the Poisson-Boltzmann equation at these interfaces to estimate the free energy of a RNA-virion. I think that the paper is well written and explores an interesting phenomenology never considered before. However, I have a number of questions/suggestions that could improve and clarify some obscure points of the manuscript before considering publication.

RESPONSE:

We are grateful to the Reviewer for her/his careful reading of the manuscript and positive view of our results. We are also grateful for the constructive criticisms, to which we provide a point-by-point reply below.

COMMENT:

In particular, the authors simplify an enveloped virus, such as SARS-CoV-2 as two concentric shells with opposed charge density for RNA (negative) and the virus lipid shell (positive). Actually, in the structure of coronavirus and other RNA viruses, such as influenza A, RNA is assembled in complex ribonucleoproteins structures far away of being a shell (Science 2012, 338 (6114), 1634-1637). In the case of the lipid shell, it is known that the lipid membrane of a cell can be understood as a capacitor whose negative and positive plates are located inside and outside the cell, respectively. However, at the end of page 1, the manuscript reads "... is adsorbed to the (positively charged) interior of the protein capsid". In particular, SARS-CoV-2 is an enveloped virus whose capsid is made of lipids. Enveloped viruses are softer than protein viruses, showing stiffness ten times lower (0.01 N/m) than protein capsids (Biophysical Journal (2011) 100(3) 637-645). In particular, flue is apparently very elastic and does not show fragile rupture under compressing forces. Therefore, it is difficult to compare with the breaking force of protein viruses as proposed by the authors (19). My question here: how can the authors justify that their model is enough to simulate enveloped viruses such as SARS-CoV-2? I think it is a bit of a stretch to pretend that this model can describe SARS-CoV-2.

RESPONSE:

We should stress here that our purpose was not to model SARS-CoV-2 specifically, but an average generic RNA virus, and we have now attempted to clarify this further. We agree that the double-shell model we use is very simplified in terms of the charge distribution. However, similarly simplified models have previously proved very useful and provide the basis for our generic understanding of the electrostatics of RNA viruses in the bulk (i.e., without any interface), see e.g. those reviewed in Ref. [9], which we used to build our theory. We have now commented on the potential importance of the detailed charge distribution for specific RNA viruses (see paragraph beginning "A limitation of our concentric shell model..." on page 2). We have also commented in the Discussion that it would be of interest in the future to study a fully 3D RNA charge distribution, as this can be quantitatively important for the specific cases of SARS-Cov-2 and the influenza virus.

Regarding the charges of inner and outer membrane, we note that the case of capsids of RNA viruses is different from that of lipid membranes of cell. In A. Bozic et al., J. Biol. Phys. 38, 657 (2012), it is shown that normally capsids of RNA viruses are mostly positively charged in their inner (hypotopal) membrane and mostly negatively charged in the outer (epitopal) membrane, whereas the overall charge is normally positive. This makes sense as the virus has to self-assemble, and an electrostatic attraction between capsid proteins and RNA will facilitate the self-assembly of functional virions. For simplicity we have not used a three-shell model (RNA + inner capsid shell + outer capsid shell) but only a two-shell model (RNA + capsid, as done to study the spontaneous assembly of virions in the bulk, see e.g. A. Siber et al., Phys. Rev. E 78, 051915 (2008)), although the two should give qualitatively similar results. As previously

mentioned, we have now commented on this limitation more in detail and we hope that this clarifies the choice of the charge distribution for our model, given the goal to study a generic case of an RNA virus.

Finally, regarding the issue of capsid breakage, we have now reworded the sentence on potential rupture, to cover also the case of softer viruses such as influenza as discussed by the Reviewer. This value is useful to compare the electrostatic forces we find with those known from viral mechanics, but our theory does not depend in any part on such a value.

COMMENT:

In page two the manuscript reads “..the dimensionless charge density σ^* ...” which depends on the charge density σ . They take σ^* to be about 17.2. How does this value fit with the charge of viruses? *Nanoscale*, 2015,7, 17289-17298. In the page three the manuscript reads “... the force resisting adsorption and associated with the electrostatic free energy barrier is $\sim 0.1 - 1$ nN for typical viral parameters.” I am not sure that this force can destroy a soft enveloped virus. Even in the case of the fragile protein viruses, 1 nN is in the low limit of their rupture force (*PNAS* (2004) 101, 7600, *PNAS* (2009) 106 (24) 9673-9678, *J Biol Chem.* 2012 Sep 7; 287(37): 31582–31595, *eLife* 2018;7:e37295. DOI, *Nanoscale* 2019 11(9):4015-4024).

RESPONSE:

The charge density is indeed taken to be representative of that in viruses. The cited work (*Nanoscale* 2015) measures charge densities for the capsids of specific viruses of about $0.1 e_0/\text{nm}^2$, fully in line with the typical values we have considered. We have now added this reference, and other relevant ones, when giving the values of charges used. We have now commented that the forces may not be sufficient for complete rupture, but only for deformation, giving most of the useful references flagged by the Reviewer as relevant citations. Nevertheless the increase in free energy is substantial, which suggests that the virion will lose stability (even if it may not rupture) at the interface. We also note that the exact value of the force (and free energy) will depend on the charge and size of the virus, and larger viruses such as influenza will experience larger forces than for a 20 nm size radius as considered in Fig. 2. The scaling of the force, as well as that of the free energy, has also been discussed in the revised version.

COMMENT:

In the description of the model (fig. 1a) I miss the separation between the shells 2δ .

RESPONSE:

We have now included the shell thickness in the schematic in Fig. 1a.

COMMENT:

It would be helpful to include the force value (N) at the right axis of the charts at figure 2.

RESPONSE:

We now include force curves for all the cases studied in Fig. 2 in the new Fig. S2. We agree these are useful to show.

Reply to Reviewer #2

COMMENT:

The authors study the interactions between viruses and interfaces using analytical theory and simulations. They find that electrostatics can explain why the presence of air-water interfaces can inactivate and kill viruses. They solve the non-linear Poisson-Boltzmann equation for a viral particle to obtain its behavior at an air-water or a liquid-liquid interface. They find that at an air-water interface the electrostatic energy and in consequence the free energy of RNA viruses increase significantly, due to the low permittivity of and the absence of electrostatic screening in air.

This is an interesting paper because of its relevance to the cleaning and disinfection of contaminated surfaces. The authors prove that electrostatics could be the underlying physical mechanism for viral inactivation at air-water interfaces. The paper is relatively well-written. However, the authors need to clarify several assumptions that they have made throughout the paper. The authors might consider the following suggestions to improve the paper:

RESPONSE:

We are very grateful to the Reviewer for her/his careful reading of and engagement with our work, and for the overall positive view that our results are interesting. We appreciate the technical questions and the constructive criticism and have addressed all the comments in our revised version. In particular we have now performed a number of additional simulations with the same electrostatic parameters for medium I and III as suggested by the Reviewer. We have also discussed the role of the background term in the free energy. Overall, the new simulations are in line with those in our original version, but we agree that this new choice of parameters is more appropriate hence we have substituted the previous results with the new ones. Below is a detailed reply to all the comments, together with a description of the corresponding changes in the revised version.

COMMENT:

1. The authors state that the electrostatic energy stored in viral capsids and genomes is estimated to be $10^4 k_{BT}$ for typical RNA viruses. It is not clear why this force is positive. Many RNA viruses assemble spontaneously both in vivo and in vitro due to the attractive electrostatics interactions. The electrostatic interaction is indeed the driving force for assembly. I agree that in the case of empty shells, the electrostatics only opposes to the assembly. However, the interaction of RNA and capsid proteins promote the assembly of virus particles. Indeed, in many cases, the viral shells do not assemble in the absence of RNA. The virus mostly assembles due to the interaction of RNA and capsid proteins. The authors need to explain this. Based on the authors discussion, the electrostatic interaction between RNA and proteins should get stronger at the water-air interface due to lack of screening,

RESPONSE:

This is a good point, which made us realise that our description of the stored electrostatic energy of the system was not sufficiently clear. The energy we compute can be thought of as the work to assemble the system from positive and negative charges at infinity. This quantity therefore does not have a simple interpretation in terms of self-assembly. A more relevant quantity to understand the self-assembly potential of the RNA+capsid in the bulk is the difference between the self-energy which we compute and, for instance, that of an empty positively charged viral capsid. This is equivalent to what done in A. Siber et al., Phys. Rev. E 78, 051915 (2008): if we did that, the difference can indeed be negative confirming that electrostatic interactions are indeed important for the self-assembly as found in that work. In our case, though, the main emphasis is to find the difference of the overall system (RNA+capsid) when it is taken to the interface. Therefore the self-energy of the assembled capsid is only a constant (the value infinitely far from the interface), and is not really important for the change in free energy. We have now: (i) rephrased our introduction so as to eliminate this confusion, and (ii) clarified our calculations after Eq. (2), which should now make it easier to understand why the electrostatic self-energy is positive.

COMMENT:

2. The authors need to explain what pickering effect is in the introduction where the first time talk about it.

RESPONSE:

Now done (in the Introduction), we agree this is useful to the reader.

COMMENT:

3. The authors assume that the number of charges on the shell is more than the number of charges on RNA. In reality, the number of charges on RNA is way more than those of viral proteins. Does a higher charge density in the inner shell have an impact on the conclusion of the paper? Can the authors solve the problem if the charge densities are different?

RESPONSE:

The problem and numerical simulations can be generalised to the case of different charge densities. The electrostatic free energy barrier would substantially increase if the RNA charge increases to be much larger than the capsid protein. However, the conclusions are unaffected, as the electrostatic energy increase would still lead to viral destabilisation when the viral particle is placed at the interface (indeed, this would be even enhanced). We have now commented on the value of calculations with different charge densities in the Discussion.

COMMENT:

4. Most viruses are permeable to salt and water. Is there a reason that the authors consider a different salt concentration inside and outside of a virus at the interface.

RESPONSE:

We wanted to consider a more general case in which the medium inside has slightly different physical properties (screening and dielectric constants) with respect to the “water” phase (see, e.g., Fig. 1 in Ref. [1]). We agree however that, in our context, it is after all a more natural assumption to say that medium I and III are in chemical equilibrium and have the same screening and dielectric constant, due to virus permeability. Therefore we have now redone the simulations with this simplified assumption, and now present the figures where medium I and III have the same electrostatic parameters. This also leads to a slightly simpler presentation. The results are qualitatively unchanged and fully in line with those in the previous version but we agree that this choice is less likely to confuse a reader. We thank the Reviewer for raising this point.

COMMENT:

5. While calculating the electrostatic interaction, do the authors consider the background energy? This is very important and can change the result of the paper. See for example the work of Podgornik and collaborators: Siber A and Podgornik R 2008 Phys. Rev. E 78 051915 and Erdemci-Tandogan G, Wagner J, van der Schoot P, Podgornik R and Zandi R 2016 Phys. Rev. E 94 022408.

RESPONSE:

This is a good point, and we are grateful to the Reviewer for raising it. We have now modified our method discussion and highlighted the contribution of the background counterion concentration in Eq. (2) and Supplementary Note 1, as this is indeed important. This is not the dominant contribution for our parameter choice, but it does need to be included. We have also mentioned the useful references highlighted by the Reviewer to signal the importance of this term – we thank the Reviewer for flagging these up.

Reply to Reviewer #3

COMMENT:

“Electrostatic inactivation of RNA viruses at air-water and liquid-liquid interfaces” is an interesting paper on a timely subject. Overall the ideas promulgated by the authors are new and reasonable but their formal development and implementation is not yet quite convincing. I would be happy to support the publication of this paper once the issues described in detail below are successfully confronted and resolved.

RESPONSE:

We thank the Reviewer for her/his careful reading of our work and for the overall positive view on it. We are also very grateful for the constructive criticism, which we have now all addressed in the revised version.

COMMENT:

-“we solve numerically the non- linear Poisson-Boltzmann (PB) equation for a viral particle approaching an air-water or a liquid-liquid interface.” There is no indication of how they calculated the electrostatic interaction free energy for the full non-linear PB theory. In fact, the electrostatic self-energy, Eq. (2), is actually valid only for the Debye-Huckel approximation and not for the full PB equation, Eq. (1). The correct form of the electrostatic free energy corresponding to the full non-linear PB equation is described in Verwey Overbeek either with Eq. 23, Eq. 26, in chapter III. If the self-energy expression Eq. 2 was used for all the results described, then they contain crucial errors as one should use the full non-linear PB theory at the specified values of the parameters. This needs to be clearly specified either in the main text of the paper or in the SI.

RESPONSE:

This is a good point and we apologise for this issue in our original version. We have now highlighted the background counterion contribution in the free energy in Eq. (2) and shown in the SI, Supplementary Note 1, where it comes from. This does not have a dominant effect in our calculations but of course we agree it definitely needs to be included. We also clarify that we solve numerically the full non-linear Poisson-Boltzmann equation.

COMMENT:

- “charged concentric spherical shells of average radius R and with spacing 2δ between them (Fig. 1a).” The figure does not indicate what is 2δ . The figure should correspond clearly with the model and should contain explicitly also the 2δ .

RESPONSE:

This detail has now been added to the schematic in Fig. 1a.

COMMENT:

- “we consider an equal charge density, σ , for both shells, so that the viral particle carries a net charge”. The charge of the virions have been measured in electrophoretic mobility experiments (see several Gelbart et al. papers) and is indeed finite, depending crucially on the pH and ionic strength of the solution. However, contrary to the stated assumption of the paper, there are indications that the inner surface charge makes no contribution to the overall charge of the virus. This is what transpires from, e.g., the AFM studies by Hernandez-Perez, et al. who found that in the case of the adenovirus the presence of its DNA did not affect the overall charge, while experiments by Johnson, et al., for both CCMV and BMV, show that the electrophoretic mobility is insensitive to the packaged RNA. Thus the assumption of the outer and inner charged shell additively contributing to the total charge of the virus in the paper cannot be clearly substantiated by experiments.

RESPONSE:

This is a good and valid point. The issue is that the apparent/effective charge of the viral particle is indeed not an additive sum of the charges, but a nonlinear function of the charges and the geometry, as there is screening in the media. This is now noted in a comment in Supplementary Note 2 after Eq. (S19). We thank the Reviewer for highlighting this useful point about the effective charge of the viruses.

COMMENT:

- “The interior of a capsid (medium III in Fig. 1) is likely a different electrostatic environment from the aqueous surrounding. We set $\kappa = 0.1\kappa$ (as the capsid may only be partially permeable

to salt) and $\epsilon_3 = \epsilon_1/16$ [9].” The medium III is usually in chemical equilibrium with medium I as the protein-based capsid is permeable to ions while the lipid membrane envelope contains proteins which can act as ion channels. The assumption that the screening in medium I and III differ thus has no ground in actual virus properties. Also, checking the cited Ref [9] confirms this statement, as that paper refers to the dielectric constant of the protein envelope and not the inside medium, which appears to be about 1/16 of the water epsilon. Also the authors do not vary the screening length inside (“The properties of the virion interior, ϵ_3 and κ_3 , also affect the results, but they are not varied below.”) so one cannot assess whether this erroneous assumption has any effect on the results or not. This needs to be checked and analysed, while the assumption of different Debye length inside and outside the virion needs to be dropped.

RESPONSE:

We thank the Reviewer for this valid observation. In our original manuscript we considered three different media for generality (this is sometimes considered in the field of viruses at surfaces, see, e.g., Fig. 1 in Ref. [1]). However we agree that in our case it is natural to consider medium I equal to medium III, for the reasons stated by the Reviewer (ion permeability of the viral capsid). We have therefore redone all our calculations with the simpler assumption that medium I and III are the same. As the results depend on the approach to the air phase at the interphase, these changes do not qualitatively alter our conclusions or results.

- the scaling on the size of the capsid, R , is also obtained purely from the linearized DH theory and can be only valid for small values of the surface charge. The scaling needs to be checked also with the full PB theory, unless it can be clearly shown that the regime under consideration fall outside its range of validity.

RESPONSE:

The scaling actually was obtained from nonlinear Poisson-Boltzmann simulations in Fig. 2A (inset). We apologise if this was unclear and we have made sure to clarify it in the revised version.

COMMENT:

- the proposed model explicitly ignores pH effects and has thus a very limited validity, as acknowledged in the MS. The conclusions are thus at best suggestive.

RESPONSE:

As the referee says, we do state this as a limitation in the manuscript. We have further discussed this simplification in the Result and Discussion section. Note that even if pH changes the quantitative value of the charge, the fact that the free energy should increase close to an interface would still hold.

COMMENT:

- while the authors state that “Electrostatics is therefore a generic physical mechanism for viral inactivation at air-water interfaces” their data (Fig. 2) and the overall description on p.3 seem to indicate rather the reverse, it seems to be the free energy associated with the surface breaching (the Pickering free energy) that favours adsorption. The electrostatics actually opposes it. And quite strongly for that matter so that without the Pickering term the virus is actually strongly repelled from the surface. In this sense the title of the paper is a bit misleading as it is the opposite, i.e., the non-electrostatic forces of the Pickering type, that appear to inactivate the virus. One might suggest the title of the paper should suggest this finding.

RESPONSE:

We see the Reviewer’s objection here. What we meant is that the electrostatic self-energy increase is a mechanism of destabilisation. It is true that in order to be subject to that increase the viral particle has to make it to the surface in the first place, and here the Pickering

contribution is needed in our theory. We have therefore reworded the title, which now reads "Mechanism for inactivation of RNA viruses at air-water and liquid-liquid interfaces". We thank the Reviewer for this suggestion and for stimulating us to rethink our title.

COMMENT:

- the discussion of the electrostatic forces indicates that they are mostly due to image interactions, either because of the discontinuity of the dielectric properties or the discontinuity in the screening parameter, both lead to a kind of dielectric images, as is known from the literature. Interestingly, the authors never mention any image effects which seem to be crucial for the resulting interactions between a virion and an interface. I think the image mechanism of the strong electrostatic repulsion needs to be clearly analyzed and discussed explicitly.

RESPONSE:

This is an interesting point. The issue is that image theory is directly applicable only to the case where there is no screening in either of the two media. For instance the works in A. Morozov et al., Phys. Rev. E 102, 020801 (R) (2020), R. R. Netz, Phys. Rev. E 60, 3174 (1999), clarify that the potential cannot be calculated with image interactions as soon as there is some screening in either medium. This is why we do not discuss image interactions. We agree that in the absence of screening image interactions would provide a very nice and natural way to quantify electrostatic energies and forces.

COMMENT:

- the authors state that "To understand these results, we formulate a Debye- Hu \ddot{c} ckel scaling theory valid for $\kappa_1 R \gg 1$ and $\delta/R \ll 1$, which is physically relevant for RNA viruses." There is no mention of the charges in this statement which are crucial for estimating the validity of the DH approximation. The authors actually admit that "quantitative predictions require full PB numerics." and it remains unclear how Eq. 4 could be used even qualitatively. The limits of validity especially those depending on the charge density should be established clearly and explicitly.

RESPONSE:

This is a good point. We discussed the value of the dimensionless charge at the beginning of the manuscript, but we have now reiterated that it is important this is small for the Debye-Hueckel theory to work. We note that our Debye-Hueckel theory is only a guide to understand qualitatively the results; we have performed the full non-linear Poisson-Boltzmann calculations and these are the data which should be looked at for quantitative predictions. These data also show the qualitative prediction of the linearised Debye-Hueckel theory are reasonable. We have now further clarified this in the Debye-Hueckel theory section.

COMMENT:

- in analyzing the disinfecting properties of ethanol the authors fail to mention the effect of ethanol on the genetical material, which is possibly much larger than the dielectric effects that they are describing. Ethanol (and other alcohols) increase the electrostatic coupling and would promote a condensation of the genome, rendering it ineffective. The authors could comment on this.

RESPONSE:

We agree, and have now commented on the effect of alcohols and ethanol on genome condensation. As discussed in our work, the effect we analyse will only be one of the reasons why ethanol is a good disinfectant.

COMMENT:

- the crucial scaling as R^2 for both the Pickering and the electrostatic parts that is required for the fine tuning the interfacial parameters needs to be viewed with caution as the linearized electrostatic theory consistent with the R^2 scaling cannot be extended outside its regime of validity. Also, the assumption of a thin RNA shell at the surface is a very idealized depiction of the RNA distribution in the real viruses. If the results described remain valid only for such a thin RNA shell they should be viewed more as an artifact of the idealized model than a real physical property of virus shells or complete virions.

RESPONSE:

As highlighted above, the scaling is actually coming from Poisson-Boltzmann simulations taking into account nonlinear effects (inset of Fig. 2A), and we hope this is clear in the new version. We have added in the Discussion the importance of going beyond the double-shell charge distribution in the future. We note though that this approximation has been successfully used in the past to study the electrostatics of RNA viruses in the bulk, without any interface. We should also add that since submitting the manuscript we have come to realise that the R^2 scaling is actually more general than we originally anticipated. For instance the case of a single shell leads to an R^3 dependence if there is air in part of the virion, but if there is screening in medium III as long as the virion is at the interface then a generalisation of the theory in Supplementary Note 3 shows the free energy gain would still scale as R^2 . This is pleasing as it renders the double-shell assumption less crucial. It remains true that for the specific case of RNA virions the scaling of electrostatic, interfacial and hydrophobic/van der Waals capsid self-assembly free energies are all the same. We have slightly reworded the discussion of our results in Fig. 3 to reflect all this (see paragraph before Discussion).

COMMENT:

- while the authors explicitly state that “Our Poisson-Boltzmann formalism takes into account the spatial charge distribution of the virion and non-linear effects due to the highly charged nature of the virion’s constituents.” the bulk of the results depend on the Debye-Huckel calculations and would probably not hold in the full non-linear PB framework. This needs to be addressed and the reader should get a clear idea what are the limitations of approximations and what are the limitations of the model.

RESPONSE:

As should now be clearer, all results in the text are actually obtained with Poisson-Boltzmann simulations, only the theory in Eq. (4) which is derived in the SI depends on the Debye-Hueckel approximation. Nevertheless we have discussed in the conclusions again the limitations of the Debye-Hueckel theory used to qualitatively explain our results.

Reviewers' Comments:

Reviewer #1:

Remarks to the Author:

The authors have responded satisfactory to my concerns, although there are still some points that have not been completely addressed.

The inactivation concept exhibited on the title is somehow blurred. The authors invoke inactivation as structure destabilization, for example in page 3 "These interactions are also similar in magnitude to the calculated electrostatic free energy increase, and we therefore hypothesize that the total energy of a virus lodged at an interface may become positive and trigger destabilization or disassembly. Electrostatics is therefore a generic physical mechanism for viral inactivation at air water interfaces."

What does a virus need to be inactivated? How much deformation or breakage is needed for inactivation? Most studies of infectivity pinpoint on the fact that viruses do not work in biochemical assays, and do not pay attention to its structural origin, unless some active biocide agent is used to block virus functionalities, such as binding, uncoating and genome translocation. Although this manuscript is putting numbers to the physical constrains underwent by viruses at the discussed interfaces, it does not provide direct proofs of how these constrains are affecting to their structure or infectivity. The title has been changed to "Mechanisms for inactivation ...", but the manuscript does not demonstrate inactivation itself. I would think more about "Physical (stress, assaults, constrains, obstacles, impediments, etc) of RNA viruses at air-water and liquid-liquid interfaces". The unresolved question is whether these physical barriers are enough for virus inactivation or not, and this probably pertains more to "Discussion" than to "Results".

In page 2 they mention "We also note that we model virions of a fixed shape, which is a good approximation until they are subjected to forces of ~ 1 nN". However, in the new figure S2A, forces are reaching until 4 nN in water. Is there a contradiction here?

Reviewer #2:

Remarks to the Author:

The authors have revised the ms based on my recommendation and have addressed all the questions/suggestions I had posed. Thus, I recommend the paper for publication now.

Reviewer #3:

Remarks to the Author:

I believe the authors have answered all my queries to an extent that I no more object to the publication of the MS. Nevertheless, I would like to make the following observations that the authors may optionally include (or not) in the final version of the paper:

The authors state that the image theory is directly applicable only to the case where there is no screening in either of the two media. For instance the works in A. Morozov et al., Phys. Rev. E 102, 020801 (R) (2020), R. R. Netz, Phys. Rev. E 60, 3174 (1999), clarify that the potential cannot be calculated with image interactions as soon as there is some screening in either medium.

This is not entirely correct as the images can originate in dielectric inhomogeneities (referred to by those two papers), or in the inhomogeneities in the screening medium (confinement of ions, spatially dependent screening as explained in Kanduc et al., Dressed Counterions: Poly- and Monovalent Ions at Charged Dielectric Interfaces, PHYSICAL REVIEW E 84, 011502 (2011).) which lead to solvation images. Both effects are non-additive and coupled.

Reply to Referee 1

COMMENT:

The authors have responded satisfactory to my concerns, although there are still some points that have not been completely addressed. The inactivation concept exhibited on the title is somehow blurred. The authors invoke inactivation as structure destabilization, for example in page 3 “These interactions are also similar in magnitude to the calculated electrostatic free energy increase, and we therefore hypothesize that the total energy of a virus lodged at an interface may become positive and trigger destabilization or disassembly. Electrostatics is therefore a generic physical mechanism for viral inactivation at air water interfaces.” What does a virus need to be inactivated? How much deformation or breakage is needed for inactivation? Most studies of infectivity pinpoint on the fact that viruses do not work in biochemical assays, and do not pay attention to its structural origin, unless some active biocide agent is used to block virus functionalities, such as binding, uncoating and genome translocation. Although this manuscript is putting numbers to the physical constrains underwent by viruses at the discussed interfaces, it does not provide direct proofs of how these constrains are affecting to their structure or infectivity. The title has been changed to “Mechanisms for inactivation ...”, but the manuscript does not demonstrate inactivation itself. I would think more about “Physical (stress, assaults, constrains, obstacles, impediments, etc) of RNA viruses at air-water and liquid-liquid interfaces”. The unresolved question is whether these physical barriers are enough for virus inactivation or not, and this probably pertains more to "Discussion" than to "Results".

RESPONSE:

We are grateful to the Reviewer for judging our response satisfactory and for her/his continuing positive view on our work. We are also grateful for the additional comments. In particular, we agree with the Reviewer that we prove destabilisation, and whether this is sufficient for inactivation is still to some extent speculative. To account for this, we have: (i) changed the title to “Mechanisms for destabilisation of RNA viruses at air-water and liquid-liquid interfaces”, and (ii) made sure to change “inactivation” to “destabilisation” when referring to our numerical and analytical results in the “Results” section, leaving the discussion that destabilisation may lead to inactivation in the “Discussion” section.

COMMENT:

In page 2 they mention “We also note that we model virions of a fixed shape, which is a good approximation until they are subjected to forces of ~ 1 nN”. However, in the new figure S2A, forces are reaching until 4nN in water. Is there a contradiction here?

RESPONSE:

The value of 1 nN is an order-of-magnitude estimate from previous work, giving an approximate idea of parameter ranges where our assumption of spherical shape may need to be revised to give more quantitatively accurate estimates. However, the main conclusion that the electrostatic free energy will increase as the virion approaches the interface does not depend on exact capsid shape, hence this limitation, noted in the current version, is not in contradiction with any of our results. Additionally, we note for capsids where 1 nN were sufficient for capsid failure, this would only reinforce our conclusion that approach to interface destabilises viral particles.

Reply to Referee 2

COMMENT:

The authors have revised the ms based on my recommendation and have addressed all the questions/suggestions I had posed. Thus, I recommend the paper for publication now.

RESPONSE:

We are very grateful to the Reviewer for the feedback that he previously gave us, as well as for recommending the current version of our manuscript for publication.

Reply to Referee 3

COMMENT:

I believe the authors have answered all my queries to an extent that I no more object to the publication of the MS. Nevertheless, I would like to make the following observations that the authors may optionally include (or not) in the final version of the paper:

The authors state that the image theory is directly applicable only to the case where there is no screening in either of the two media. For instance the works in A. Morozov et al., Phys. Rev. E 102, 020801 (R) (2020), R. R. Netz, Phys. Rev. E 60, 3174 (1999), clarify that the potential cannot be calculated with image interactions as soon as there is some screening in either medium.

This is not entirely correct as the images can originate in dielectric inhomogeneities (referred to by those two papers), or in the inhomogeneities in the screening medium (confinement of ions, spatially dependent screening as explained in Kanduc et al., Dressed Counterions: Poly- and Monovalent Ions at Charged Dielectric Interfaces, PHYSICAL REVIEW E 84, 011502 (2011).) which lead to solvation images. Both effects are non-additive and coupled.

RESPONSE:

We thank the Reviewer for agreeing to the publication of our work, and for the additional comment on the image charges. We agree that accounting for screened, dielectric and ionic cloud images as in the work cited by the Reviewer will allow an alternative way to study our system. We have now mentioned this paper in the Discussion, as it would be useful in the future to generalise our study to the case where there are also polyvalent counterions in the contacting media I and/or II.